# Metformin Protects against Radiation-Induced Acute Effects by Limiting Senescence of Bronchial-Epithelial Cells

**DOI:** 10.3390/ijms22137064

**Published:** 2021-06-30

**Authors:** Christine Hansel, Samantha Barr, Alina V. Schemann, Kirsten Lauber, Julia Hess, Kristian Unger, Horst Zitzelsberger, Verena Jendrossek, Diana Klein

**Affiliations:** 1Institute of Cell Biology (Cancer Research), University Hospital, Essen, University of Duisburg-Essen, 45122 Essen, Germany; Christine.Hansel@uk-essen.de (C.H.); sjbarr@wisc.edu (S.B.); Alina.Schemann@gmail.com (A.V.S.); Verena.Jendrossek@uk-essen.de (V.J.); 2Department of Radiation Oncology, University Hospital, LMU München, 80539 Munich, Germany; Kirsten.Lauber@med.uni-muenchen.de; 3German Cancer Consortium (DKTK), Partner Site Munich, 80539 Munich, Germany; 4Clinical Cooperation Group ‘Personalized Radiotherapy in Head and Neck Cancer’ Helmholtz Zentrum München, German Research Center for Environmental Health GmbH, 85764 Neuherberg, Germany; julia.hess@helmholtz-muenchen.de (J.H.); unger@helmholtz-muenchen.de (K.U.); zitzelsberger@helmholtz-muenchen.de (H.Z.); 5Research Unit Radiation Cytogenetics, Helmholtz Zentrum München, German Research Center for Environmental Health GmbH, 85764 Neuherberg, Germany

**Keywords:** radiotherapy, normal tissue toxicity, lung injury, pulmonary disease, premature senescence, senescence-associated secretory phenotype, metformin

## Abstract

Radiation-induced damage to normal lung parenchyma remains a dose-limiting factor in thorax-associated radiotherapy (RT). Severe early and late complications with lungs can increase the risk of morbidity in cancer patients after RT. Herein, senescence of lung epithelial cells following RT-induced cellular stress, or more precisely the respective altered secretory profile, the senescence-associated secretory phenotype (SASP), was suggested as a central process for the initiation and progression of pneumonitis and pulmonary fibrosis. We previously reported that abrogation of certain aspects of the secretome of senescent lung cells, in particular, signaling inhibition of the SASP-factor Ccl2/Mcp1 mediated radioprotection especially by limiting endothelial dysfunction. Here, we investigated the therapeutic potential of a combined metformin treatment to protect normal lung tissue from RT-induced senescence and associated lung injury using a preclinical mouse model of radiation-induced pneumopathy. Metformin treatment efficiently limited RT-induced senescence and SASP expression levels, thereby limiting vascular dysfunctions, namely increased vascular permeability associated with increased extravasation of circulating immune and tumor cells early after irradiation (acute effects). Complementary in vitro studies using normal lung epithelial cell lines confirmed the senescence-limiting effect of metformin following RT finally resulting in radioprotection, while fostering RT-induced cellular stress of cultured malignant epithelial cells accounting for radiosensitization. The radioprotective action of metformin for normal lung tissue without simultaneous protection or preferable radiosensitization of tumor tissue might increase tumor control probabilities and survival because higher radiation doses could be used.

## 1. Introduction

Patients receiving thoracic irradiation for the treatment of lung, breast, or hematologic malignancies are prone to suffer from normal tissue toxicity [1]. Radiation-induced normal lung damage limits the use of maximum radiation doses that could be delivered aiming at achieving high tumor control probabilities, as lungs are quite radiosensitive organs [2,3]. Summarized as radiation-induced lung injury (RILI), radiation-induced pneumonitis (acute phase) and pulmonary fibrosis (late phase), are the two dose-limiting toxicities following indicated radiotherapy (RT) treatments [4,5]. According to the time following RT, the complex radiation-induced changes involving numerous resident cell types within the lung, including parenchymal (epithelial), mesenchymal, and vascular cells together with infiltrating immune cells, can be divided into different phases [5]. Within hours and the first few days, direct DNA damage or damage to cellular components following the generation of reactive oxygen species (ROS) impacts on gene expression profiles and levels, including the release of cytokines and growth factors, such as interleukin-1 (IL-1) and IL-6, tumor necrosis factor-α (TNF-α), platelet-derived growth factor-β (PDGF-β), and basic fibroblastic growth factor (bFGF). Further on, leukocyte infiltration, pneumocytes type I apoptosis, and intra-alveolar edema become obvious. 6–8 weeks following RT, alterations in lung perfusion, hypoxia, and increased expression of additional cytokines (e.g., transforming growth factor-beta 1 (TGFβ1)) gain up [5]. The clinical RT-induced pneumonitis phase manifests 3–12 weeks after RT and comprises loss of epithelial and endothelial barrier function (alveolar collapse) on the one hand, as well as vascular endothelial growth factor (VEGF)-mediated endothelial cell proliferation and pneumocyte type II proliferation for re-epithelializing the alveolar basement membrane on the other hand [3,5]. Prior to the fibrotic phase, which can appear after six months, the pre-fibrotic or intermediate phase is characterized by extracellular matrix (ECM) generation from activated fibroblasts that migrate and proliferate in the alveolar walls. Herein, pro-fibrotic TGFβ1 is supposed to be decisive for the excess production of ECM and the differentiation of fibroblasts into myofibroblasts finally resulting in increasing thickening and stiffness of the lung parenchyma [4,5]. An improved understanding of the underlying mechanisms of RILI is prerequisite to provide mechanistic bases for targeted therapies and to investigate potential radioprotective treatment strategies that can rationally minimize radiation-induced pneumonitis and/or pulmonary fibrosis and therefore maximize a survivor’s quality of life.

A central response following diverse stressors, including ionizing radiation results in physiological dysfunction, particularly in lungs is cellular senescence [6,7,8,9]. This permanent cell cycle arrest bears marked morphological as well as characteristic gene expression alterations. High cytokine and growth factor expression and secretion by senescent cells, known as the senescence-associated secretory phenotype (SASP), that were associated with persistent DNA damage signals, could reinforce the cell cycle arrest, and alter the cell’s microenvironment with regard to the stromal composition and the stimulation of immune responses [10,11,12]. Among the different types of lung cells, senescent bronchial and alveolar epithelial cells, fibroblasts, and endothelial cells, have been reported to prevail in diseased lungs, with a supposed central role of epithelial senescence [8]. Particularly found with RT-induced senescence of bronchial epithelial cells and respectively up-regulation of the senescence-associated secretory phenotype (SASP) factor, chemokine C-C motif ligand 2 (Ccl2), has led to the activation of the hitherto quiescent healthy endothelium, resulting in increased endothelial permeability associated with increased leakage of blood stream components into the lung interstitium, which then fosters inflammation and/or metastasis formation [13,14]. Of note, therapeutically applied mesenchymal stem cells during the early phase after thoracic irradiation efficiently counteracted epithelial senescence as well as vascular dysfunction [13,14]. Furthermore, signaling inhibition of the SASP-factor Ccl2 resulted in radioprotection especially of the vascular compartment that coincided with reduced inflammation within the acute phase and limited fibrosis outcome [15].

Agents that modulate the expression of constituents of the SASP, so-called SASP inhibitors or senomorphic drugs, have the potential to alleviate aging and associated diseased phenotypes, including cellular senescence [16]. Among these agents is metformin. Metformin is a U.S. Food and Drug Administration (Silver Spring, MD, USA) approved drug for adults with type 2 diabetes, usually medicated to decrease hepatic glucose production and glucose absorption by the gastrointestinal tract as well as to increase insulin sensitivity [17,18]. Metformin has been shown to be a potent complex I inhibitor of the mitochondrial electron transport chain, thereby suppressing the endogenous mitochondrial ROS production. Inhibition of mitochondria and decreased ROS levels in turn impact on cell type decisions. Moreover, due to its anti-inflammatory action metformin was shown to exert additional beneficial effects. In particular, the multiple inflammatory cytokines being induced and subsequently orchestrating (during) cellular senescence were known to be modulated by metformin treatment, as metformin was shown to inhibit (the NF-ꓗκB-dependent) SASP [19]. Besides the favorable influence of metformin on oxidative stress and inflammation, metformin was even shown to alleviate the aging-associated cellular senescence [20].

Here, the impact of metformin on modulating senescence induction for normal lung toxicity following RT was investigated using a well-established preclinical model for normal lung damage, the mouse model of IR-induced pneumopathy. Radioprotection of the normal tissue via senescence and/or SASP-related signaling modulation without simultaneous protection or preferable radiosensitization of tumor tissue have important implications in oncology because higher doses of radiation might improve both local control and survival.

## 2. Results

### 2.1. Metformin Treatment Prevents Radiation-Induced Senescence of Bronchial–Alveolar Epithelial Cells and Associated Vascular Dysfunction

The impact of metformin treatment on radiation-induced senescence in normal lung tissue was investigated using our mouse model of radiation-induced pneumopathy (Figure 1). C57BL/6 received a 0Gy (sham-treatment) or 15Gy whole thorax irradiation (WTI) in combination with daily metformin treatment or vehicle control via intraperitoneal injection within the first three weeks post-irradiation (Figure 1A). Senescence-associated beta-galactosidase (SA-betagal) activity staining was performed using frozen lung tissue sections at 21 days post-irradiation (Figure 1B). As expected, increased SA-betagal activities were detected in epithelial cells of larger bronchi in lungs after WTI compared to control lung sections. Of note, metformin treatment severely limited RT-induced senescence induction. Induction of senescence was further confirmed by quantifying the cellular senescence mediator cyclin-dependent kinase inhibitor 1 (Cdkn1a/p21) on protein level by Western blot and on mRNA level by qRT-PCR (Figure 1C,D) as well as the cell cycle effector *Ccnd1* (*cyclinD1*) (Figure 1D). *Cdkn1a* expression levels were significantly increased in lungs of irradiated animals and restored upon metformin treatment. qRT-PCR quantifications of *Ccnd1* mRNA expression levels in total lung RNA isolates followed similar alterations (Figure 1C).

Concerning potentially affected SASP factors, the angiogenic SASP factors *Ccl2* (*Mcp1*) and *Mmp2* (*matrix metalloproteinase-2*) were analyzed, both were shown to be critically involved in RT-mediated endothelial activation (Figure 1E). Upon WTI, both expression levels were upregulated. Metformin treatment efficiently decreased expression levels of *Ccl2* after WTI, while there were slight but significant increases in *Mmp2* expression levels. Similar trends were detected for mRNA expression levels of the immune-related SASP factors *Ccl7*, *Ccl8*, *Cxcl1* (*C-X-C motif chemokine 1*) and *Il6* compared to respective *Ccl2* expressions of control and metformin treated WTI mice. Whereas RT-induced increases of *Cxcl1*, *Il6,* and *Ccl7* were only partially normalized following the combined metformin treatment, *Ccl8* expression levels were significantly decreased. Thus, metformin treatment significantly reduced WTI-induced senescence of bronchial–alveolar epithelial cells. Furthermore, metformin impacts on certain aspects of the SASP, particularly by reducing WTI-induced Ccl2 levels, which in turn impact on adjacent vascular structures resulting in vascular dysfunction, namely increased vascular permeability, inflammation, and extravasation of circulating tumor cells as already shown [13,14,15].

Therefore, radiation-induced vascular dysfunction was investigated using an Evans-blue dye (EBD) extravasation assay (Figure 2A). Importantly, RT-induced impairments of vascular function was significantly restored in metformin treated WTI mice, as determined by increased EBD extravasation following WTI into the lung interstitium and significantly reduced EBD levels in combination with metformin treatment at 21 days post RT. Flow cytometric analysis of crude cell extracts of freshly isolated lung tissues was further performed to specify endothelial cell alterations (Figure 2B) and potentially associated immune cell infiltrations (Figure 2C). The vascular system was analyzed using antibodies against the endothelial cell surface proteins CD31 (PECAM1), CD34 and VegfR2 within the CD45-negative cell population. CD31-positive endothelial cell numbers showed no significant difference between irradiated and non-irradiated lung tissues, although endothelial cell numbers seemed to be slightly decreased (Figure 2B). Treatment with metformin did not further affect the proportion of endothelial cells within lung tissue. However, expression of CD34 and VegfR2 on endothelial cells was slightly upregulated upon WTI, indicating an activated, “angiogenic” endothelial cell phenotype.

Further, it was investigated whether the associated recruitment of myeloid cells was affected by metformin treatment (Figure 2C). No changes in CD45-positive cell numbers were detected. The composition of the CD45-positive myeloid cell compartment was further specified using CD11b and CD11c antibodies. WTI triggered an early loss of alveolar macrophages (autofluorescent CD11c (high)CD11b(int)) and a potentially compensatory influx of CD11b+ cells compared to sham irradiated controls (Figure 2C). Of note, metformin treatment resulted in a further increase of the percentage of CD11b-positive cells, more specifically of CD11b(high)CD11c(int) phagocytic cells, which bear the potential to govern the initiation of inflammation resolution phase upon metformin treatment in combination with RT (Figure 2C). WTI-induced alterations within lung tissues in combination with (Ccl2-mediated) vascular dysfunction, particularly increased vascular permeability, were shown to increase the risk of circulating tumor cell extravasation followed by metastatic seeding into irradiated lungs tissues [13,14,15].

To investigate a potential protective effect of metformin treatment on RT-induced metastatic niche, we investigated seeding and growth of intravenously injected syngeneic tumor cells in previously irradiated lung tissue (Figure 3). Highly metastatic B16F10 melanoma cells were injected into the tail vein at 21 days post-irradiation and metformin treatment (Figure 3A). Tumor cell extravasation and (subsequent) metastasis formation was significantly increased in irradiated lungs 14 days after tumor cell injection, whereas metformin treatment significantly reduced seeding of circulating tumor cells and subsequent metastasis formation (Figure 3B,C).

Conclusively, metformin treatment efficiently limited WTI-induced senescence induction of bronchial–alveolar epithelial cells and aspects of its associated SASP, namely increased Ccl2 levels. Most likely thereby, metformin treatment counteracted WTI-induced vascular dysfunction, associated immune cell infiltrations, and metastatic seeding of circulating tumor cells.

### 2.2. Metformin Treatment Prevents RT-Induced Senescence of Cultured Normal Bronchial–Alveolar Epithelial Cells Finally Resulting in Radioprotection, While Fostering RT-Induced Cellular Stress of Cultured Malignant Epithelial Cells Indicating Radiosensitization

Next, complementary in vitro investigations were performed in order to prove the beneficial effect of metformin on irradiated normal lung epithelial cells, and to reveal potential differences in the radiation response of normal lung epithelial cells compared to malignant epithelial cells, namely non-small cell lung cancer (NSCLC). Two lung epithelial cell lines, BEAS-2B (bronchial epithelial cells) and HSAEC1-KT (small airway epithelial cells) established from normal lung epithelium, were used in combination with high dose radiation (15 Gy; according to the in vivo examinations) and metformin treatment (2 and 10 mM). Respective cultures were firstly analyzed regarding alterations in cell cycle, apoptosis, and cell death levels (Figure 4).

Cell cycle analysis of BEAS-2B cells revealed irradiation effects as a significant decrease of cells in G_1_ phase, a slight increase of cells in G_2_/M phase, a significant induction of SubG_1_ levels as well as an increase of cells with a higher DNA content (>4n phase) at 96 h post RT (Figure 4A and Appendix A). While metformin treatment resulted in a slight increase of cells in G_1_ phase in a concentration dependent manner (48% at 0 mM to 53% at 10 mM) in non-irradiated cells, a more pronounced but not significant increase in numbers of cells in G_2_/M phase were detected following combined metformin treatment in a concentration dependent manner (from 27% at 0 mM to 37% at 10 mM) upon irradiation with 15 Gy. DNA fragmentation, depicting apoptosis as revealed in increasing SubG_1_ levels, showed an irradiation dependent, significant increase of SubG_1_ levels, while no additional metformin effect was detected (Figure 4B). Respective long-term treatments further excluded possible harmful effects of metformin (0–10 mM) when combined with RT (0–6 Gy) as revealed by clonogenic survival analysis (Figure 4C). Corresponding analyses were performed with HSAEC1-KT cells yielding similar results (Figure 4D–F and Appendix A). Again, metformin treatment resulted in a slight increase of cells in G_1_ phase in a concentration dependent manner (50% at 0 mM to 56% at 10 mM) in non-irradiated cells, while a significant increase in numbers of cells in G_2_/M phase were detected following combined metformin treatment in a concentration dependent manner (from 27% at 0 mM to 47% at 10 mM) upon RT. Of note, using higher metformin concentrations (10 mM) combined with RT seemed rather protective for HSAEC1-KT cells, as indicated by lower apoptosis levels (Figure 4E), but considering that the plating efficiency was also drastically reduced here, which in turn limited colony formation (Figure 4F). The combined treatment of RT with metformin at a concentration of 2 mM significantly increased clonogenic survival (Figure 4F). Likewise, epithelial cell viability and proliferation was significantly reduced following RT, whereas metformin treatment only impacted on decreased cell viability by tendency, an effect that was less pronounced when metformin treatment was combined with radiation (Figure 4G,I). Increasing metformin concentrations however resulted in reduced proliferation rates, without further limitations for the combined treatment (Figure 4H,J).

Conforming to in vivo examinations, the influence of metformin on radiation-induced senescence was analyzed in vitro using the cultured non-malignant epithelial lung cells (Figure 5). Protein expressions of the tumor suppressor (p53)-p21-dependent senescence pathway, and the survival marker protein kinase B (PKB/AKT) were determined by Western blot analyses in lysates of control or metformin treated cultures in combination with RT (Figure 5A,E). p21 expression levels were clearly induced upon irradiation in both cell lines, 96 h post RT. Metformin treatment decreased RT induced p21 expressions in a concentration dependent manner. p53 phosphorylation levels seemed to decrease slightly upon radiation treatment, and metformin treatment further reduced these levels. The survival protein AKT seemed to be active in both epithelial cell lines, and metformin either applied alone or in combination with RT downregulated these levels (Figure 5A,E). RT induced senescence formation was specified in BEAS-2B and HSAEC1-KT epithelial lung cells using flow cytometry in combination with the fluorescent dye C12FDG, which can be hydrolyzed by β-galactosidase in enriched lysosomes upon senescence yielding in green fluorescence (Figure 5B,F). Both, lung epithelial cells showed a similar extent of induced senescence formation following RT with 15 Gy. Of note, metformin treatment significantly decreased senescence formation in irradiated BEAS-2B cells and by tendency in HSAEC1-KT. Lower RT doses (6 Gy) resulted in reduced senescence formation (around 2–5%; not shown). Corroborating experiments were performed using the classic SA-betagal, staining followed by quantification of senescent cells by counting (Figure 5C,D,G,H). Senescent cells, classically detected via blue staining resulting from cleavage of the substrate X-Gal by SA-betagal, were enriched upon cellular senescence. However, the number of SA-betagal positive HSAEC1-KT cells following radiation were reduced compared to BEAS-2B epithelial cells, which correlates with in vivo findings where RT-induced senescence mainly affected epithelial cells of larger bronchial structures.

As mitochondrial respiration alterations following metformin treatment might be due to metformin’s complex I inhibitory effect, and the mitochondrion and more precisely its respiratory chain is the main sources of ROS, which in turn is the main source of oxidative damages, RT-induced mitochondrial ROS production following RT stress was analyzed in combination with metformin treatment (Figure 6). ROS levels of non-malignant lung epithelial cells were analyzed according to the metabolic measurements, 24 h post treatment, using Dihydroethidium staining via flow cytometry. RT-induced ROS production was more pronounced in BEAS-2B compared to HSAEC1-KT (Figure 6A,C). Of note, whereas metformin treatment did not affect ROS levels in non-irradiated cultures, metformin treatment limited ROS formation in both cell lines in combination with RT (15 Gy). While 10 mM metformin treatment significantly reduced ROS formation in high dose irradiated BEAS-2B cells, the decrease in ROS following metformin treatment alone was only detected by tendency. A similar trend was shown for HSAEC1-KT cells, as metformin treatment did not impact on ROS levels itself but reduced ROS levels in irradiated HSAEC1-KT cells (Figure 6C). As it was already shown for the 96 h timepoint following RT, radiation also increased the number of cells in G_2_/M phase, in BEAS-2B (from 25% to 50.4%) and HSAEC1-KT cells (from 24.7% to 50.4%), 24 h post RT (Figure 6B,D). SubG_1_ levels were not affected by combined treatment and thus apoptosis induction was unaltered at that time point.

Further on, the impact of metformin in metabolic reprogramming during RT-induced senescence was analyzed by extracellular flux measurements (Figure 7). Mitochondrial function was analyzed upon inhibition of ATP synthesis by oligomycin, upon uncoupling of oxidative phosphorylation with FCCP, and upon blocking mitochondrial respiration by using rotenone and antimycin A. RT significantly impacted on mitochondrial respiration in bronchial epithelial cells (BEAS-2B) and in small airway epithelial cells (HSAEC1-KT), 24 h post RT (Figure 7A,I). Metformin treatment (alone) significantly reduced mitochondrial respiration in both cell types, effects that were even more pronounced compared to RT (Figure 7B,J). Combining RT then with metformin treatment had only a slight additional effect on epithelial lung cells as compared to metformin alone, indicating that metformin was the main actor in mitochondrial respiration reduction (Figure 7C–E,K–M). Within basal respiration, general energy demands of non-malignant lung cells were unraveled. Both cell lines depicted similar levels here (Figure 7D,L) that were significantly reduced upon RT, further reduced by metformin treatment alone or when combined with RT. A similar trend was observed for maximal respiration rates (Figure 7E,M). Likewise, oxygen-linked ATP productions were significantly reduced upon RT, even further upon metformin as well as combined metformin treatment in BEAS-2B and in HSAEC1-KT (Figure 7F,N). Mitochondria efficiencies as indicated by the lack of proton leak increases following the different treatments were confirmed in both lung epithelial cell lines (Figure 7G,O). At the same time, decreases in proton leak as energetic change were detected following RT and/or metformin treatment as proton leak is a highly significant aspect of mitochondrial energetics and is implicated in oxygen consumption. Confirmatory, a significant increase in glycolysis in BEAS-2B and an increase by tendency in HSAEC1-KT was determined following metformin treatment through an increase of the extracellular acidification rates (ECAR) of the surrounding media, which is predominately from the excretion of lactic acid per unit time after its conversion from pyruvate (Figure 7H,P). Following RT and the combination of RT with metformin however reduced glycolysis rates in both cell types and metformin reduced the flexibility of glycolysis induction following ATP synthase (complex V) inhibition by oligomycin that was shown to decrease the electron flow through the electron transport chain and thus reducing mitochondrial respiration, indicating a general decrease of metabolic demands.

Conclusively, radiation-induced cell death levels as estimated for two different epithelial cell lines established from normal lung epithelium were not affected by metformin treatment and did not affect respective clonogenic survivals rates. At the same time, metformin treatment limited RT-induced senescence in both epithelial cell types. Thus, metformin treatment of normal epithelial cells did not exert adverse effects but could rather act protective against RT-induced complications. The early changes in the metabolic profile coincided with the potential change in the cellular phenotype, particularly proliferation versus cell cycle arrest at G_2_ following radiation-induced senescence induction. Herein, metformin reduced cellular stress, particularly when combined with RT, finally leading to a quiescence-like stage, a metabolically specific state characterized by suppressed catabolism without necessitating cell death.

To further exclude tumor-promoting features of metformin treatment when combined with RT as well as to further specify if and how metformin impacts on the RT response of malignant epithelial cells compared to normal lung epithelial cells, respective in vitro investigations were performed using the non-small cell lung cancer cells NCI-H460 (human epithelial large cell lung carcinoma) and A549 (human alveolar basal epithelial adenocarcinoma). Cell viabilities were nearly unaffected by RT or metformin treatment but seemed to reduce upon increasing metformin concentrations (10 mM) when combined with RT (Figure 8A,E). RT as well as metformin treatment significantly reduced both NSCLC cells’ proliferation (Figure 8B,F). The decrease in proliferation was accompanied with significantly increased numbers of NCI-H460 and A549 cells in G_2_/M cell cycle phases (from 10% to 38% and from 18% to 38%, respectively) upon RT accounting for cell cycle arrest, with further slight increases (up to 45%) upon metformin treatment (Figure 8C,G and Appendix A). Concerning the G_1_ levels, radiation significantly reduced cell numbers in the G_1_ phase of both malignant epithelial cell lines. Metformin treatment significantly decreased (non-irradiated) NSCLC cells in the G_1_ phase at higher concentrations (10 mM), while no additional effect of combined treatment was detected. At the same time, SubG_1_ levels increased upon RT in NCI-H460 and by tendency in A549 cells, with A549 cells being more radioresistant as indicated by low levels of apoptosis following RT with no obvious influence of metformin treatment. Total cell death levels were increased upon RT and even slightly more upon combined treatment (Figure 8D,H). Of note, cell death rates even seemed to increase following metformin treatment in non-irradiated cells. Estimation of clonogenic survival rates further confirmed the difference in radioresistance (Figure 8I). Furthermore, metformin treatment dramatically reduced the plating efficiency of both NSCLC cells in a concentration-dependent manner, finally resulting in vanishing colony numbers.

Corresponding metabolic analysis was performed as described above for the lung epithelial cell lines established from normal lung epithelium (Figure 9). Upon irradiation, both cell lines showed only minimal impairments in mitochondrial respiration or upon metformin treatment (Figure 9A,B,F,G). RT significantly reduced basal respiration in both cell lines with a higher degree in reduction in A549 cells (Figure 9C,H), while maximal respiration was only reduced by tendency. Compared to non-malignant epithelial cells (Figure 6), NSCLC cells’ glycolytic and oxidative potentials were strongly increased (Figure 9A,E,F,J), particularly as indicated by higher ATP synthesis levels (Figure 9D,I), which mainly rely on oxidative phosphorylation and not on glycolysis as ECAR remains unchanged after RT, metformin treatment or a combination of both were increased (Figure 9E,J). Compared to NCI-H460, the more radio-resistant A549 cells even had higher ATP synthesis levels (Figure 9D,I). Here, RT induced a significant synthesis inhibition, an effect that was less efficient in NCI-H460 cells upon combined metformin treatment, but further reduced in A549 cells indicating a heightened vulnerability of oxidative phosphorylation metabolism in A549 cells following metformin treatment (Figure 9D,I). Metformin treatment alone was not sufficient to reduce ATP levels in both non-irradiated cell types, and only slight increases in glycolysis were detected (Figure 9D,E,I,J).

Of note, ROS levels were increased upon irradiation, and importantly increased further, at least more likely upon metformin treatment (Figure 10A,E). At this time point, RT resulted in more cells in cell cycle G_2_/M phase in a dose dependent manner (from 29% to 47%) for NCI-H460 and for A549 cells (from 24% to 46%), and metformin treatment did not further affect these cell cycle distributions (Figure 10B,F). SubG_1_ levels as indicators for apoptosis were slightly increased upon irradiation at that time point, while metformin treatment did not impact on these levels (Figure 10C,G). Overall, radiation-induced cell death levels increased in a dose dependent manner with metformin being able to cause a further increase in death levels in a concentration dependent manner (Figure 10D,H).

In summary, compared to normal epithelial cells, both NSCLC cells revealed clearly higher oxidative phosphorylation levels and extracellular acidification rates, thus an enhanced metabolism for the increased energy requirements resulting from increased proliferation rates, and presumably increased migration as well as invasion capacities (Appendix A). Whereas RT-induced stress, as revealed by increasing ROS levels, were accompanied by senescence induction in normal epithelial cells, and efficiently counteracted upon metformin treatment, RT-induced ROS production in malignant epithelial cells was further increased upon combined metformin treatment finally increasing cell death levels, and thus indicating radiosensitization.

## 3. Discussion

Modulation of the radiation response of normal tissues aiming at achieving radioprotection is a central goal of radiobiology. Following thoracic irradiations, senescence induction turned out to be a key pathway orchestrating several alterations within the lung’s microenvironment [8,21]. Accumulation of senescent cells in turn can contribute to radiation-induced lung injury onsets and impact on pneumonitis as well as fibrosis progression, a large variety of cell and non-cell autonomous effects [8,9]. Thus potential, pharmacological strategies aiming at improved senescent cell clearance and/or limiting the extent of senescence induction and associated phenotypes represent promising therapeutically options. Within that scenario we previously reported that abrogation of certain aspects of the secretome of irradiated resident lung cells, in particular by inhibiting signaling of the SASP factor Ccl2 derived from RT-induced senescent bronchial epithelial cells, mediated radioprotection of normal lung tissue especially with respect to the vascular compartment [14,15].

Using the well-established preclinical model of RT-induced pneumopathy we have shown, that the antidiabetic drug metformin successfully limited the induction of senescence in lungs following thoracic irradiations, which beneficially affected the lung’s microenvironment, particularly resulting in restored vascular dysfunctions. Within the acute phase following RT generally endothelial activation could be observed, so switching the quiescent phenotype towards a pro-inflammatory one, an endothelial state that is associated with endothelial dysfunction [22,23]. Consequently, increased vascular permeability associated with edema and inflammation, deterioration of the vascular tone and blood hemostasis problems gain up. The combined application of metformin with RT efficiently counteracted the increased vascular permeability following RT as well as the associated increase in extravasation of circulating immune and tumor cells. Of note, these radioprotective actions were accompanied by reduced levels of the angiogenic SASP factor Ccl2 following metformin treatment.

Metformin was already shown to be able to reduce pro-inflammatory cytokine secretion, particularly Ccl2 [24]. It has been supposed that metformin can limit the onset and duration of a cytokine storm-like responses, particularly upon lung damage thereby preventing excessive lung damage [25,26,27]. For example, metformin efficiently counteracted the lipopolysaccharide-induced inflammatory response within acute lung injury [28,29].

Since most of the SASP factors, in particular Ccl2, impact on immune cell recruitment, we analyzed whether RT-induced alterations in the lung secretome in combination with metformin treatment resulted in alterations of lung infiltrations of myeloid cells. Although no changes in CD45 positive cell numbers could be reported following WTI, an early loss of pulmonary macrophages became prominent associated with a potentially compensatory influx of CD11b(+) cells, as this myeloid cell population is known to contain macrophage precursors with the supposed potential to replace eradicated alveolar macrophages [30,31]. In addition, the number of CD11b(high)CD11c(int) cells were significantly increased in WTI lungs following metformin treatment, which bear the potential to govern the initiation of inflammation resolution phase. CD11b(high)CD11c(int) phagocytic cells, as previously identified by the additional expression of MHC II and CD64, were designated as CD11b(high) interstitial macrophages and could, due to their potential immunoregulatory properties, be considered as a second line of defense by expressing immunosuppressive cytokines [32,33]. Accordingly, elevated cell numbers of CD11b(high)CD11c(int) cells infiltrating lungs when RT was combined with metformin indicate that these cells might contribute to the restoration of the homeostatic microenvironment following RT upon metformin treatment. However, it has to be stated that the impact of metformin for RT-induced immune alterations was rather low.

RT-induced Ccl2 expressions in lungs were already linked to the recruitment of myeloid cells with reported tumor-promoting activities [13,14,15]. Metastatic cancer cells in turn could co-opt chemokine pathways to foster metastatic seeding at secondary sites, particularly by using factors that normally impact on the recruitment of immune cells [34]. We showed here that metformin when combined with RT, successfully reduced the extravasation and subsequent formation of metastases, which is in line with previous reports stating that metformin was able to reduce metastases numbers without affecting tumor growth [35]. With respect to potential immune cell alterations, metformin was shown to act beneficially on inflammation by inducing M1 macrophage polarization towards a more anti-inflammatory M2 phenotype, thus promoting anti-inflammatory pathways [35,36].

With respect to the vascular compartment, metformin treatment was even shown to exert anti-inflammatory and antioxidant effects [37,38]. Elevated levels of oxidative stress in leukocytes of diabetes patients were reported together with increased levels of endothelial adhesion molecules that in turn caused increased leukocyte–endothelial interactions, effects that were restored in metformin-treated patients [38]. Accordingly, lower levels of circulating proinflammatory cytokines were estimated in older patients suffering from adult-onset diabetes upon metformin treatment [39]. Likewise, metformin was shown to improve mitochondrial function and vascular homeostasis in aortic tissue and microvascular endothelial cells by decreasing ROS levels from mitochondria and NADPH oxidase and thus increasing the endothelial antioxidant capacity [40,41]. At the same time, metformin treatment did not impact on endothelial cell cytotoxicity, but decreased induced cell proliferation and migration, and reduced MMP expression levels, indicating quiescence promoting features [42]. Inhibiting angiogenesis or limiting endothelial activation could thus be another rationale, whereby metformin exerts its protective effects concerning normal tissue, while the same properties of metformin could account for its anti-cancer effects [43].

Thus, RT-induced lung damage that usually is associated with damage to vascular structures as reported here, finally resulting in increased permeability and associated extravasation of blood stream components, can be efficiently limited by counteracting RT-induced senescence and associated SASP when combining RT with metformin treatment. However, metformin treatment within the early phase after irradiation (within three weeks post RT) does not prevent from chronic senescence, nor radiation-induced endothelial cell loss and fibrosis as adverse late effects at 25 weeks post RT (not shown), which strongly argues for a constitutive metformin application as most frequently applied for patients. Using another preclinical mouse model of RT-induced pneumopathy (chest irradiation of rats with 18Gy using a cobalt-60 gamma ray source) it was already shown that metformin mitigated pulmonary fibrosis [44]. Herein, fibrosis development was completely reversed following metformin treatment (at 100 days post RT). The attenuation of the continuous production of free radicals (chronic ROS production) from mitochondria and stimulation of antioxidant defense enzymes or DNA repair enzymes was suggested to mediate the protective action by metformin [44,45,46,47]. The beneficial effects on fibrosis was further strengthened, when vascular and alveolar damages were attenuated upon metformin treatment in the same rat model, which additionally revealed reduced interleukin (IL)-4 cytokine, IL-4 receptor-a1, and dual oxidase 2 (DUOX2) levels and thereby a reduced DUOX2 regulated redox-signaling following a combined metformin treatment [48].

With complementary in vitro studies using established epithelial cell lines from normal lung epithelium, we confirmed that metformin treatment is able to prevent senescence induction following RT without impacting on additional cell fate decision, namely potentially increasing RT-induced cell death levels or reducing respective clonogenic survivals. Mechanistically, metformin impacted on the AKT-p53/p21 axis for senescence induction, by reducing respective protein levels. It is well known that p53 is a critical regulator in determining cellular fates [49]. Upon increased cellular stress, including RT-induced cellular oxidative stress, and subsequent DNA damage repair induction, activated p53 can impact on p21, which bears the capacity to promote senescence through apoptosis inhibition [49,50]. p53 itself could even directly impact on the cell’s fate towards a senescence phenotype as mediated by pro-senescence acetylation of p53 while pro-apoptotic phosphorylation was inhibited [49,51]. Senescence induction following p53 activation and subsequent expression of p21 was further shown to depend on AKT/PKB [52]; p53-dependent senescence induction turned out to require a cooperation between p21 and AKT [53]. Chronic PI3K/AKT/mTORC1 pathway activation was shown to induce proliferative arrest termed AKT-induced senescence when hyperactivation of AKT promoted enhanced mTORC1-dependent p53 synthesis, resulting in a transcriptionally upregulation of p21 [51]. The SASP in turn seems to depend on AKT activation since AKT inhibition could reduce the associated SASP factors [53]. On the other hand, the pro-survival factor AKT/PKB acts to maintain a regular mitochondrial potential and thus, regular and efficient ATP levels. Likewise, p53 has emerged as potential regulator of mitochondrial respiration and glycolysis, including ROS production, by its ability to increase aerobic respiration while decreasing glycolytic pathways [54]. Accordingly, we showed here that metformin-induced reduction of the p53/p21 axis in non-malignant epithelial cells caused a down-regulation of the metabolic epithelial cell profile, which is predominately orchestrated by mitochondrial respiration without an obvious impact on glycolysis. Treatment with metformin alone was already shown to reduce ATP-linked respiration, maximal respiratory capacity, and basal respiration [55]. Consequently, RT-induced ROS levels in normal lung epithelial cells were significantly reduced upon metformin treatment. ROS accumulation in turn could foster abnormalities in mitochondrial morphology and function [56]. Thus, targeting mitochondria and mitochondria-dependent processes comprise a potential strategy for mitigation of radiation-induced lung injury, particularly by attenuating initiation of inflammation as well as fibrotic pathways, by a combined treatment with metformin.

As another important aspect, metformin could improve the antioxidant defense of cells by increasing the activity of altered antioxidant enzymes such as superoxide dismutase, catalase, and glutathione, together with the ability to enhance DNA repair capacity [57,58]. These normal tissue modulating actions of metformin finally resulting in radioprotection, particularly concerning RILI, could even be modulated and importantly sensitize cancer cells to radiotherapy [58,59]. Accordingly, combining metformin treatment with RT was successfully shown to mediate anti-cancer properties, namely radiosensitization [60,61,62,63]. Herein, radiosensitization by metformin has been attributed to the PI3K/AKT/mTOR pathway or through improved oxygenation, reduces tumor hypoxia [63]. Of note, while metformin increased the radiosensitivity of cancer cells by increasing intracellular ROS levels, the effect on basal phenotype cells was negligible [64]. Consistently, we showed here that NSCLC cell viabilities, namely of NCI-H460 and A549 cells following RT, were reduced upon co-treatment with increasing metformin concentrations. Together with the decrease in cellular proliferations and accompanied G_2_/M arrests, the clonogenic survival rates were significantly impaired. Tumor cells, as characterized by higher proliferation rates, are supposed to be more sensitive to metformin treatment, which is based on their activation of AMP-activated protein kinase (AMPK). For example, it was shown that inhibition of oxidative phosphorylation (by complex 1 inhibitory function of metformin) and a respective decrease in cellular ATP levels led to the activation of the intracellular energy sensor AMPK that activated specific enzymes and growth control nodes to increase ATP generation and to decrease ATP consumption [65]. Whereas in normal epithelial cells, mitochondrial oxidative phosphorylation is primarily efficient in generating ATP for energy homeostasis, the increasing energy demands of malignant epithelial cells are accomplished by increased mitochondrial respiration as well as glycolysis as determined by increased mitochondrial respiration and glycolysis rates. Many cancer cells strongly rely on fermentative glycolysis, known as the Warburg effect, while expressing increased factors that aim to reduce oxidative stress. Targeting the enhanced aerobic glycolysis predominating in many cancer cells could be a potential therapeutic strategy to reduce tumor burden [66]. Accordingly, the glucose metabolism affecting metformin was already shown to exert anti-tumor properties, although the underlying molecular mechanism has not been fully clarified. Importantly, it was suggested that tumor cells, which have a high reliance on oxidative phosphorylation as a source of ATP and concurrently lack metabolic flexibility to efficiently engage glycolysis, would be most susceptible to metformin’s action [67]. In line with the presented results, both investigated NSCLC cell types were shown to mainly rely on oxidative phosphorylation and not on glycolysis after RT, metformin treatment or a combination of both. Consequently, non-malignant epithelial lung cells turned out to be more resistant to ATP-inhibition following metformin treatment most likely due to the lower steady state levels, whereas malignant epithelial cells, that highly proliferate and need more ATP, preferentially undergo cell death upon metformin treatment, and thus ATP-inhibition, resulting in increased cellular stress. Accordingly, increased ROS levels following metformin treatment (in combination with RT) were estimated in both NSCLC cells used here. ROS levels in turn critically regulate apoptotic cell fate decisions [68,69]. Furthermore, targeted induction of ROS in cancer cells emerged as a potential approach in cancer therapy [70]. Metformin was already shown to successfully prevent cancer cell viability and cell proliferation by inducing G_2_/M cell cycle arrest in a dose and time-dependent fashion followed by programmed cell death induction, effects that were attributed to increased ROS levels [68]. Likewise metformin targeted mitochondria-dependent apoptosis, and inducing DNA breaks by activating ROS [71]. Moreover, ROS levels vary during cell cycle progression and peak in mitosis. Prolonged mitotic arrest then additionally increased ROS levels and thus increased levels of oxidatively damaged biomolecules that accumulate [72]. Mitotic arrest agents might further enhance the effects of ROS-dependent cancer therapies. In line with these findings, RT induced G_2_/M arrests in both NCI-H460 and A549 cells, an effect that was more pronounced in cancer cells compared to normal epithelial cells, where senescence was induced following G_2_/M arrest after RT. In contrast to the growth arrest known as quiescence, which classically happens in G_0_, growth arrest resulting in senescence could occur in the G_1_ and possibly in the G_2_/M phase of the cell cycle [73].

Taken together, cancer cells were more sensitive to starvation-induced cytotoxicity and metabolic oxidative stress than non-transformed human cell types, strongly suggesting metformin as a potential agent in cancer therapy to increase the therapeutic window [74,75,76]. Whereas RT-induced stress, as revealed by increasing ROS levels, was accompanied by senescence induction in normal epithelial cells, and efficiently counteracted upon metformin treatment, RT-induced ROS production in malignant epithelial cells was further increased upon combined metformin treatment, indicating radiosensitization. Consequently, both cell fates, apoptosis, and senescence following cellular stress and particularly following RT-mediated oxidative stress, were shown to rely on the (cellular) proliferation state. Thus, metformin is beneficial for preventing normal lung damage when combined with RT, particularly for preventing RT-induced senescence thereby mediating (radio) protection, especially of the vascular compartment without simultaneous protection of respective cancer tissues and rather radiosensitizing the latter one.

*Limitations:* For diabetic patients, maximum metformin doses of 2250 mg per day (about 30 mg/kg) result in serum concentrations out of 10–20 μM [77,78]. Concerning the in vitro studies, substantially higher and potentially physiologically irrelevant doses (2-10 mM) were established, as cultured cells were usually exposed to unusually high levels of nutrients and growth factors as media supplements that could limit metformin treatment efficiency [78]. Thus, the in vitro used millimolar metformin concentrations may not be efficient for predicting concentrations required for metformin effects in vivo or in clinical treatments [77]. However, it is important to determine metformin preclinical concentrations in animal models in order to determine if these levels could be achieved in patients [77]. Therefore, the preclinical metformin dose of 125 mg/kg per day (applied via intraperitoneal injection) as used here seemed relatively high compared to an average of 30 mg/kg per day orally administered in anti-diabetic treatment schedules for humans. Considering the shorter half-life of metformin in mice, together with an interspecies scaling factor of 12.3, as provided by the US Food and Drug Administration and the National Cancer Institute, the metformin concentrations applied to mice are far below the human maximum safe dose (42.5 mg/kg) [79,80,81]. Although within therapeutic metformin doses, genotoxicity or mutagenicity issues could be excluded [82], adverse drug reactions particularly metformin related side effects to digestive tract symptoms and/or lactic acidosis were known to appear rather frequently [83]. In patients, metformin-related gastrointestinal adverse reactions increased as the number of six risk factors (initial dose of 750 mg, female, age ≤65, body mass index ≥25, aspartate aminotransferase ≥30 IU/L and alkaline phosphatase ≥270 IU/L), strongly suggesting that optimization of the metformin dose would be beneficial for patients with type 2 diabetes mellitus [83]. Metformin at very low concentrations (≤50 µM) for example, was already shown to be efficient in reducing RT-induced damage in human lymphocytes [84]. And currently, on-going clinical trials are investigating the combined use of daily metformin with RT, including low-dose metformin [58,59].

## 4. Materials and Methods

### 4.1. Whole Thorax Irradiation (WTI)

All animal experiments were performed in accordance with ethical approval of the local Committee on the Ethics of Animal Experiments of the Landesamt für Natur, Umwelt und Verbraucherschutz (LANUV), Regierungspraesidium Duesseldorf. C57BL/6 (wildtype, WT) mice were bread and housed under specific pathogen free conditions at 12 h dark light cycles in the central animal facilities of the university hospital in Essen. C57BL/6 mice (mixed gender) received 15 Gy of WTI in a single dose as previously described [13,14,15]. For combined metformin treatment, mice were daily (five-times a week) treated with metformin (Sigma—Aldrich, St. Louis, MO) by intraperitoneal injection of 125 mg/g metformin dissolved in PBS or PBS alone (vehicle control) starting two hours prior to WTI. Intraperitoneally administration was chosen because metformin was shown to be effective than when given orally, most likely due to a higher plasma level of metformin after injection than oral administration, at least in preclinical models [85,86]. According to the principles of the 3Rs (replacement, reduction and refinement), stress for the animals through intraperitoneally injections was considered to be lower than with (daily) oral gavages. Three weeks post WTI, animals were narcotized using isoflurane and killed by transcardial perfusion. Lungs were isolated and directly subjected for PFA-fixation, RNA or protein isolation (fresh frozen material in liquid nitrogen) or FACS analysis. For vascular leakage Evans blue dye (EBD) (Sigma-Aldrich) extravasation from the blood stream into the lung interstitium was analyzed by an intravenous injection of 100 µg EBD/100 µL PBS around two hours prior to sacrificing respective anesthetized animals by transcardial perfusion with PBS to remove vascular blood. EBD extraction was performed as previously described [13,15]. For tumor cell extravasation and subsequent metastasis analysis, 0.5x 10^6^ highly metastatic B16F10 tumor cells were intravenously transplanted 21 days after WTI and metformin treatment. Additional 14 days later, lungs were subjected for immunohistochemistry as previously described [13,15].

### 4.2. Senescence-Associated Beta-Galactosidase (SA-Betagal) Activity

SA-betagal activity was detected at pH6 using frozen lung tissue sections [13,15]. In brief, slides were washed twice in PBS, incubated with X-gal staining solution (0.1% X-gal, 5 mM potassium ferrocyanide, 5 mM potassium ferricyanide, 150 mM NaCl, and 2 mM MgCl_2_ in 40 mM citric acid/sodium phosphate solution, pH 6; all chemicals were from Sigma—Aldrich) for 12–16 h at 37 °C, and washed with PBS prior embedding.

### 4.3. Flow Cytometry

Immune cells were infiltrated upon WTI into lung and vascular markers and analyzed by flow cytometry of crude cell extracts from freshly isolated cells as previously described [15,87]. Isolated cells were stained with fixable viability dye eFluor780 (APC Cy7, eBioscience Inc., San Diego, CA, USA), with fluorochrome-labeled anti-mouse CD45 (30-F11, BioLegend, San Diego, CA, USA) to identify living leukocytes. Cells were further stained with CD31 (FITC), CD34 (APC), VegfR2 (PE) and CD11b (PerCP-Cy5.5), CD11c (APC) antibodies (all from BioLegend Inc., San Diego, CA). Flow cytometric measurements were performed on a BD LSRII flow cytometer using FACS DIVA software (BD Bioscience, Franklin Lakes, NJ). For cell cultures, cell cycle, cell death rates, and ROS levels were analyzed at the indicated time point post RT using respectively harvested cells (by trypsinization) in combination with staining solution. Cell cycle analysis was performed using Nicoletti staining with 0.1% sodium citrate (Merck KGaA, Darmstadt, Germany), 0,1% Triton X-100 (Sigma—Aldrich), and 50 µg/mL propidium iodide (PI, Sigma –Aldrich) in PBS for 30 min incubation at room temperature prior analyses. Cell death levels were evaluated by PI exclusion staining using 10 µg PI/mL (in PBS). ROS levels were determined by 5 µM dihydroethidium (DHE; Sigma—Aldrich) staining (in PBS). Senescence formation was analyzed eight days post RT following bafilomycin A1 (100 nM, Biozol, Eching, Germany) and C12FDG (5-dodecanoylaminofluorescein di-β-D-galactopyranoside; 33 µM; Thermo Fischer Scientific Waltham, MA, USA) incubations.

### 4.4. Western Blot

Protein samples (50–100 µg total protein) were subjected to SDS-PAGE electrophoresis and Western blots were done as previously described [13,14,15]. p21 (F8, sc271610) antibody was from Santa Cruz Biotechnology (Dallas, TX) and phospho-p53 (Ser392); #9281, (total) p53 (1C12; #2524), Akt (#9272) antibodies were from Cell Signaling Technology (Danvers, MA). Unless otherwise indicated, representative blots from three different experiments using the indicated antibodies were shown. Respective signals were related to beta-actin (AC-74, A2228, Sigma—Aldrich).

### 4.5. Real-Time Reverse Transcription PCR (qRT-PCR)

RNeasy Mini Kit (Qiagen, Venlo, Netherlands) was used to isolate total RNA and RNA (1 µg) was reverse transcribed using Superscript^TM^-II reverse transcriptase (Qiagen), according to each manufacturer’s instructions. qRT-PCR was carried out as described before using the oligonucleotide primers: bActin-fw GGCTGTATTCCCCTCCATCG, bActin_bw CCAGTTGGTAACAATGCCATGT, Cdkn1a_fw GAGAACGGTGGAACTTTGACTT, Cdkn1a_bw CTCAGACACCAGAGtGCAAGAC, Ccl2_fw CTGCTACtCATTCACCAGCAAG, Ccl2_bw AATGTATGTCTGGACCCATTCC, Ccl7_fw ATCCACATGCTGCTATGTCAAG, Ccl7_bw GGAGTTGGGGTTTTXATGTCTA, Ccl8_fw CTACGCAGTGCTTCtTTGCC, Ccl8_bw ACATACCCTGCTTGGTCTGG, Ccnd1_fw GAGGTCTGTGAGGAGCAGAAGT, Ccnd1_bw AGGAAGTGTTCGATG-AAATCGT, Cxcl1_fw AAACCGAAGTCATCGCCACACT, Cxcl1_bw CGTTACTTGGGGACACCTTTTA, Il6_fw AGAGGATACCACTCCCAACAGA, Il6_bw CTGAAGGA-CTCTGGCTTTGTCT, Mmp2_fw GCTCCACCACATACAACTTTGA, Mmp2_bw TCGGGACAGAATCCATACTTCT), in a AriaMx Real-Time PCR System (Agilent Technologies, Santa Clara, CA) [13,14,15]. Expression levels were normalized to the reference gene (beta actin; set as 1) and were shown as relative quantification.

### 4.6. Cell Cultures

The NSCLC cell lines NCI-H460 and A549 (both from ATCC; Manassas, VA) were cultured in RPMI 1640 and DMEM media (Gibco, Thermo Fisher Scientific, Waltham, MA, USA) supplemented with 10% fetal calf serum (FCS) and 100 U Penicillin/Streptomycin (Sigma—Aldrich, St. Louis, MO, USA). BEAS-2B and HSAEC1-KT (both from ATCC) were cultured in airway epithelial cell growth media (C-21060; PromoCell GmbH, Heidelberg, Germany). Cells were cultured under standard cell culture conditions at 37 °C and 5% CO_2_. All cells were routinely tested for mycoplasma contamination (every two weeks) and periodic authenticated by STR profiling (if necessary, no later than yearly). Cells were plated and adherent cells were treated with metformin (0–10 mM in PBS) or vehicle control. Two hours later, cells were irradiated with indicated doses using a X-RAD 320 machine (Precision X-Ray Inc., North Branford, CT, USA) with 320 kV, 10 mA, and 1.65 mm aluminum-filter, at a distance of 50 cm (dose rate of 2.6–2.7 Gy/min) at room temperature. For cell viability measurements, WST-1 (water soluble tetrazolium salt; Sigma—Aldrich) was added (1/10) to respective cultures and colorimetric changes were measured at 450 nm (optical density; OD) at the indicated time points. Cellular proliferations were determined by crystal violet (CV) staining (0.1% CV in PBS; Carl Rot) of PBS washed and glutaraldehyde-fixed (0.1% in PBS; Carl Roth, Karlsruhe, Germany) cells. Following Triton X-100 CV release dye concentration were measured spectrophotometrically at 540 nm. For clonogenic survival were plated in six-well plates as previously described [15,88]. After an additional 10 days of treatment, cells were fixed and subsequently stained with 0.05% Coomassie Brilliant Blue. Colonies (≥50 cells/colony) were counted and survival curves were generated by plotting the log of the surviving fraction against the treatment dose.

### 4.7. Extracellular Flux Analysis

Cells were plated at in XF96 microplates (Seahorse Bioscience/Agilent Technologies, Santa Clara, CA) according to the manufacture’s instruction and as previously described [89]. For Mito Stress Test (Seahorse Bioscience), 24 h post treatment (with the optimized metformin concentration of 2.5 mM for this assay and 10Gy RT), medium was changed to XF base medium, supplemented with 1 mM pyruvate, 2 mM glutamine and 10 mM glucose, and incubated for 1 h at 37 °C in a CO_2_-free incubator. Mitochondrial oxidative phosphorylation on the basis of the oxygen consumption rate (OCR) and glycolysis by analyzing the extracellular acidification rate (ECAR) were estimated following oligomycin (1 µM), carbonyl-cyanide-p-trifluoromethoxyphenylhydrazone (FCCP) (BEAS-2B, HSAEC1-KT, A549: 1 µM; NCI-H460: 2 µM), and rotenone (0.5 µM) and antimycin A (0.5 µM) treatment at indicated time points using a Seahorse XFe 96 Analyzer. Hoechst 33342 (10 µg/mL, Thermo Fisher Scientific; Waltham, MA) was used for individual normalization to DNA. Data were analyzed using Wave 2.6 software (Seahorse Bioscience).

### 4.8. Statistics

If not otherwise indicated (*n* = biological replicates), data were obtained from at least 3 independent experiments. Data were presented as mean values ±SEM. Individual mice numbers (*n*) were also indicated in the respective figure legends. Data analysis was performed by one- or two-way ANOVA followed by indicated multiple comparison post-test or by unpaired (two-tailed) *t*-test using Prism7 (GraphPad, La Jolla, San Diego, CA, USA). Statistical significance was set at the level of *p* ≤ 0.05.

## Figures and Tables

**Figure 1 ijms-22-07064-f001:**
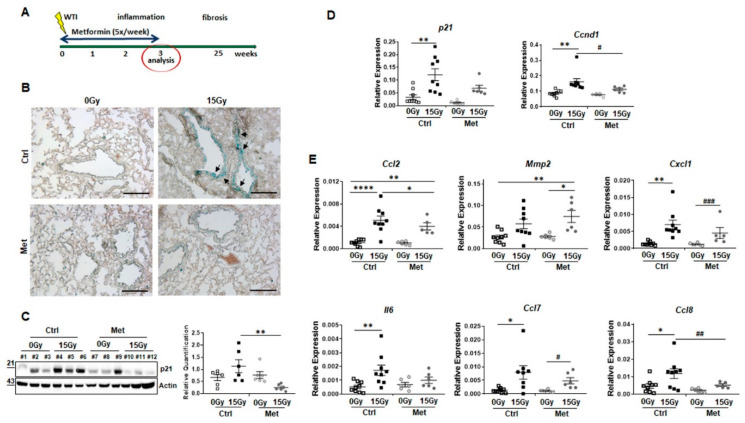
Metformin treatment limits senescence of bronchial–alveolar epithelial cells and associated senescence associated secretory phenotype in whole thorax irradiated mice. (**A**) C57BL/6 mice were left untreated (Ctrl), received five-times a week metformin (Met) treatment, 15Gy whole thorax irradiation (WTI) or received WTI in combination five-times a week metformin treatment for first three weeks post WTI. (**B**) Senescence-associated beta-galactosidase activity was analyzed on frozen lung sections, 21 days post irradiation. Photomicrographs depict representative pictures (scale bar = 200 µm). (**C**) Senescence mediator cyclin-dependent kinase inhibitor 1 (Cdkn1a/p21) protein expressions were analyzed in whole lung protein lysates using Western Blot analyses Expression levels were quantified by densitometry and data are presented as mean values ±SEM of individual biological replicates (*n* = 6–9 per group). *p*-values indicate: ** *p* ≤ 0.01 by one-way ANOVA with post hoc Tukey’s multiple comparison test. (**D**) *Cdkn1a/p21* as well as *cyclin D1* (*Ccnd1*) mRNA expression levels were further quantified using Real-Time RT-PCR. Relative transcript levels were normalized to beta-actin (set as 1). Symbols depict individual biological replicates measured in duplicates each (0Gy and 15Gy Ctrl: *n* = 9 per group; 0Gy Met, 15Gy Met: *n* = 6 per group). Data represent mean values ±SEM, *p*-values indicate: * *p* ≤ 0.05, ** *p* ≤ 0.01 by one-way ANOVA with post hoc Tukey’s multiple comparison test. Additional unpaired (two-tailed) *t*-test analysis is depicted by # *p* ≤ 0.05. (**E**) SASP factor (*Ccl2*, *Mmp2*, *Cxcl1, Il6*, *Ccl7*, *Ccl8*) expression levels were analyzed (in duplicates) on mRNA level of whole lung RNA isolates (0Gy and 15Gy Ctrl: *n* = 9 per group; 0Gy Met, 15Gy Met: *n* = 6 per group). Individual biological replicates and mean values ±SEM are shown. *p*-values indicate: * *p* ≤ 0.05, ** *p* ≤ 0.01, **** *p* ≤ 0.0001 by one-way ANOVA with post hoc Tukey’s multiple comparison test and additionally by unpaired (two-tailed) *t*-tests as depicted by ## *p* ≤ 0.01, ### *p* ≤ 0.001.

**Figure 2 ijms-22-07064-f002:**
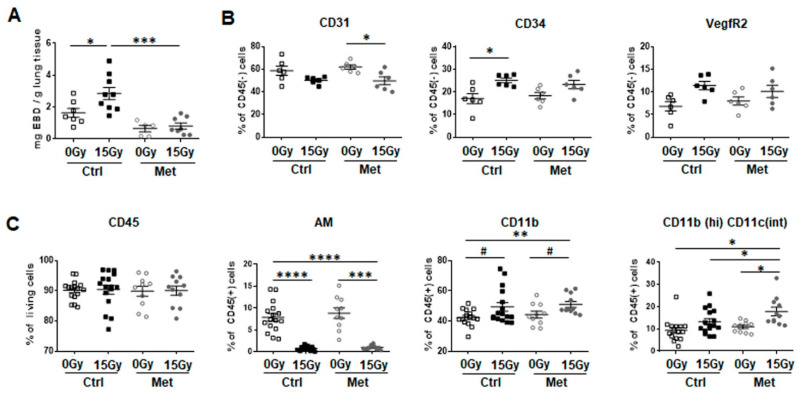
Metformin treatment counteracts radiation-induced vascular dysfunction (acute effect) and recruitment of inflammatory myeloid cells. C57BL/6 mice were left untreated (Ctrl), received five-times a week metformin (Met) treatment, 15Gy WTI, or received WTI in combination five-times a week metformin treatment for first three weeks post WTI. (**A**) Three weeks post irradiation, vascular leakage was determined by intravenously injected Evans Blue dye (EBD) extravasation from the blood stream to the lung interstitium. Extracted EBD concentrations were quantified by absorption measurements and related to lung weight. Symbols represent biological replicates (0Gy Ctrl: *n* = 7; 15Gy Ctrl: *n* = 9; 0Gy Met: *n* = 5, 15Gy Met: *n* = 8) with indicated mean values ±SEM. *p*-values indicate * *p* ≤ 0.05, *** *p* ≤ 0.001 by one-way ANOVA with post hoc Tukey’s multiple comparison test. (**B**) Endothelial marker expression in crude cell extracts of freshly isolated lung tissue were identified using CD31/PECAM1, CD34 and VEGFR2 expressions (of CD45 negative cells) by flow cytometry. Symbols depict individual biological replicates (*n* = 6 for each group) with indicated mean values ±SEM. *p*-values indicate: * *p* ≤ 0.05 by one-way ANOVA with post hoc Tukey’s multiple comparison test. (**C**) Leukocytes were identified using CD45 expression and myeloid cells were further characterized using CD11b and CD11c antibodies. Alveolar macrophages (AM) measured as CD11c(high)CD11b(intermediate) cells, CD11b(+)CD11c(-) myeloid cells (monocytes/granulocytes) and CD11b(high)CD11c(intermediate) phagocytic cells are depicted as symbols (individual biological replicates) with mean values ±SEM. *p*-values indicate: * *p* ≤ 0.05, ** *p* ≤ 0.01, *** *p* ≤ 0.001, **** *p* ≤ 0.0001 by one-way ANOVA with post hoc Tukey’s multiple comparison test and additionally by unpaired (two-tailed) *t*-tests depicted as # *p* ≤ 0.05.

**Figure 3 ijms-22-07064-f003:**
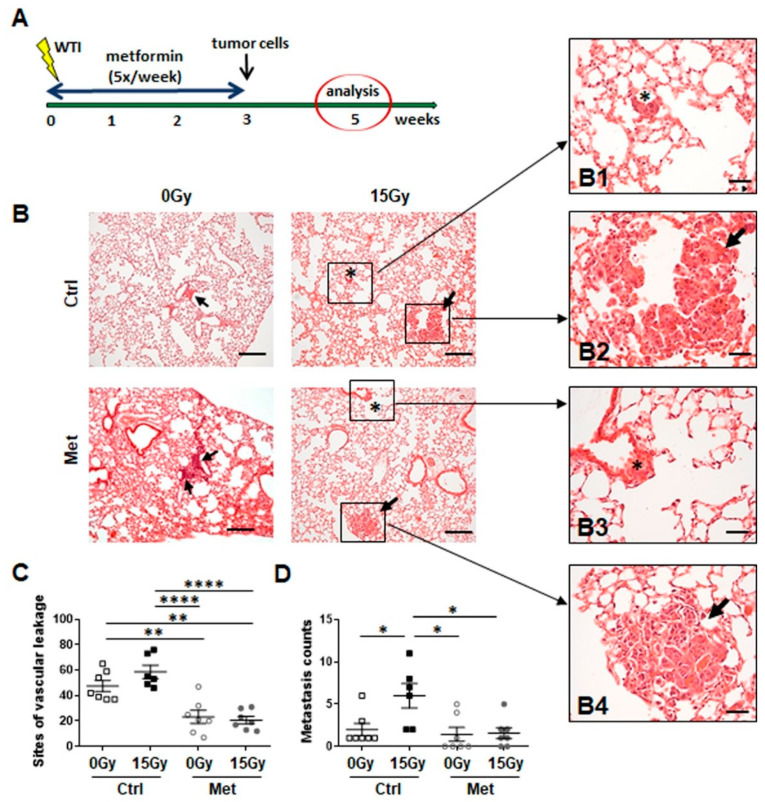
Metformin treatment limits metastasis formation in previously irradiated lung tissue. (**A**) C57BL/6 mice were left untreated (Ctrl), received metformin (Met) treatment, 15Gy WTI or received WTI in combination with metformin (Met) treatment for first three weeks post WTI. (**B**) Seeding of circulating tumor cells into the lungs was initiated 21 days after irradiation. Therefore, 0.5 × 10(6) metastatic tumor cells were intravenously transplanted via the tail vein of C57BL/6 mice. Fourteen days after tumor cell injection animals were sacrificed, lungs were isolated and subjected for lung histopathology using hematoxylin and eosin staining. Areas of extravasated tumor cells were emphasized by asterisks (B1, B3). Arrows point towards formed metastases (B1, B4). Scale bar: 200 µm; magnifications: 40 µm. (**C**) Sites of vascular leakage (designated as sites of tumor cell extravasation/micrometastasis) and (**D**) subsequently formed macrometastasis were quantified in whole lung sections (0Gy Ctrl, 0Gy Met, 15Gy Met: *n* = 7 per group; 15Gy ctrl: *n* = 6). Data are shown as individual biological replicates with mean values ±SEM. *p*-values indicate: * *p* ≤ 0.05, ** *p* ≤ 0.01, **** *p* ≤ 0.0001 by one-way ANOVA with post hoc Tukey’s multiple comparison test.

**Figure 4 ijms-22-07064-f004:**
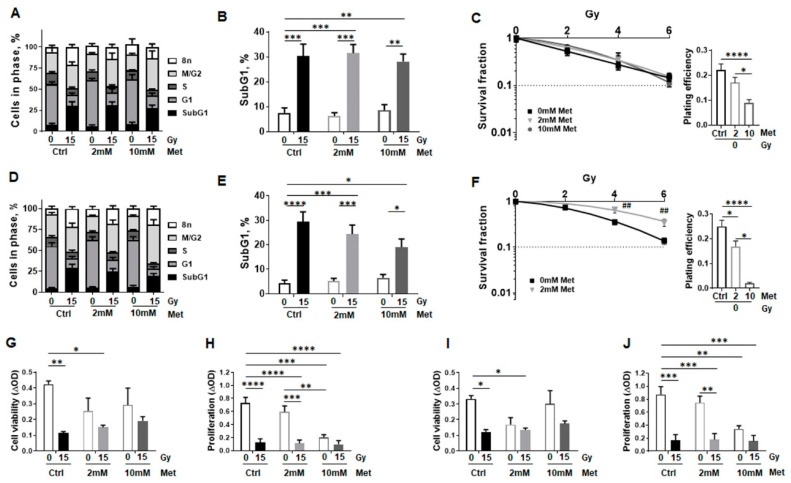
Metformin treatment does not impact on proliferation and survival of normal epithelial lung cell cultures following RT. Lung epithelial cell lines derived from normal lung epithelium (BEAS-2B: human bronchial epithelial cells, (**A**–**C**); HSAEC1-KT: human small airway epithelial cells, (**D**–**F**)) were cultured in normal growth media supplemented with metformin (2 and 10 mM) or vehicle control (Ctrl) 2 h prior radiation treatment with 0Gy or 15Gy. Cell cycle phases (**A**,**D**) and apoptotic cells (SubG1) (**B**,**E**) were analyzed by flow cytometry 96 h after irradiation. Graphs consist of data from 3–5 individual experiments (with SEM). *p*-values indicate: * *p* ≤ 0.05, ** *p* ≤ 0.01, *** *p* ≤ 0.001, **** *p* ≤ 0.0001 by two-way ANOVA with Tukey’s multiple comparison test. (**C**,**F**) Clonogenic survival was evaluated in metformin treated (0, 2, 10 mM) and irradiated (0, 2, 4, 6 Gy) in BEAS-2B and HSAEC1-KT cells at 10 days post RT. Coomassie stained colonies were quantified by counting. Data show the surviving fractions (SF) from 3 independent experiments (means ±SEM) plated in triplicates each. *p*-values indicate: * *p* ≤ 0.05, by one-way ANOVA with Tukey’s multiple comparison test. Cell viability was analyzed using WST-1 assay in BEAS-2B (**G**) and HSAEC1-KT (**I**) cells at 96 h post treatment. Cell proliferation was further determined using crystal violet staining (**H**,**J**). Data represent mean values ±SEM from 4–5 independent experiments measured in seven replicates each. *p*-values indicate: ** *p* ≤ 0.01, *** *p* ≤ 0.001, **** *p* ≤ 0.0001 by two-way ANOVA with Tukey’s multiple comparison test.

**Figure 5 ijms-22-07064-f005:**
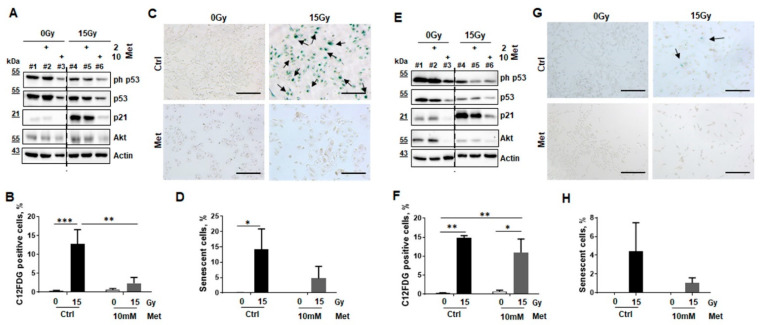
Metformin treatment prevents RT-induced senescence in normal epithelial lung cell cultures. The impact of metformin treatment for RT-induced senescence formation was investigated in cultures of lung epithelial cell lines derived from normal lung epithelium (BEAS-2B and HSAEC1-KT) by analyzing protein expression levels of senescence markers via Western blot (**A**,**E**), C12FDG staining and flow cytometry (**B**,**F**), and SA-betagal activity staining (**C**,**D**,**G**,**H**). p21 and p53 expression levels and respective levels of the cell survival marker AKT/PKB were analyzed by Western Blots at 96 h post RT and treatment in BEAS-2B (**A**) and HSAEC1-KT (**E**) cells. Representative blots of 3 independent experiments are shown. Dashed lines indicate different areas of the same blot. Radiation-induced senescence formation was analyzed by C12FDG treatment prior flow cytometric analyses in BEAS-2B (**B**) and HSAEC1-KT (**F**), eight days post treatment. Data showed mean values ±SEM of six independent experiments. *p*-values indicate: * *p* ≤ 0.05, ** *p* ≤ 0.01, *** *p* ≤ 0.001 by two-way ANOVA with post hoc Tukey’s multiple comparison test. Following SA-betagal activity staining, senescent BEAS-2B (**C**) and HSAEC1-KT (**G**) cells were visualized in in blue. Representative pictures are shown (scale bar: 200 µm). (**D**,**H**) Senescence formation was quantified by counting the numbers of SA-betagal-positive and -negative epithelial cells. Data showed mean values ±SEM of 3–5 independent experiments. *p*-value indicates: * *p* ≤ 0.05 by two-way ANOVA with Tukey’s multiple comparison test.

**Figure 6 ijms-22-07064-f006:**
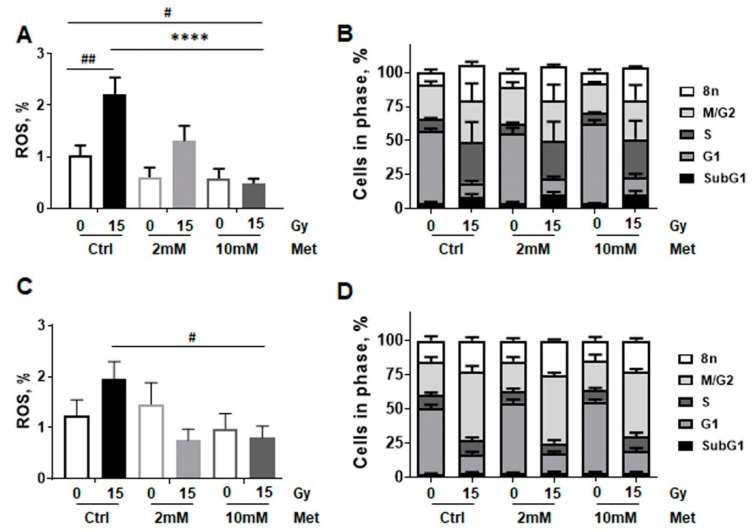
RT-induced cellular stress is reduced upon combined metformin treatment in epithelial cells established from normal lung epithelium. BEAS-2B (**A,B**) and HSAEC1-KT (**C**,**D**) lung epithelial cells were analyzed concerning reactive oxygen species (ROS) production (**A**,**C**) at 24 h of post RT and metformin (Met) treatment using dihydroethidium staining in combination with flow cytometry analyses. Data are shown as mean values ±SEM of 5–7 independent experiments. *p*-value indicates: **** *p* ≤ 0.0001 by two-way ANOVA with post hoc Tukey’s multiple comparison test and additionally by unpaired (two-tailed) *t*-tests depicted as # *p* ≤ 0.05, ## *p* ≤ 0.01. Respective cell cycle analysis (**B**,**D**) at that time point were estimated following Nicoletti staining. Data are summarized as mean values ±SEM of 3–5 independent experiments.

**Figure 7 ijms-22-07064-f007:**
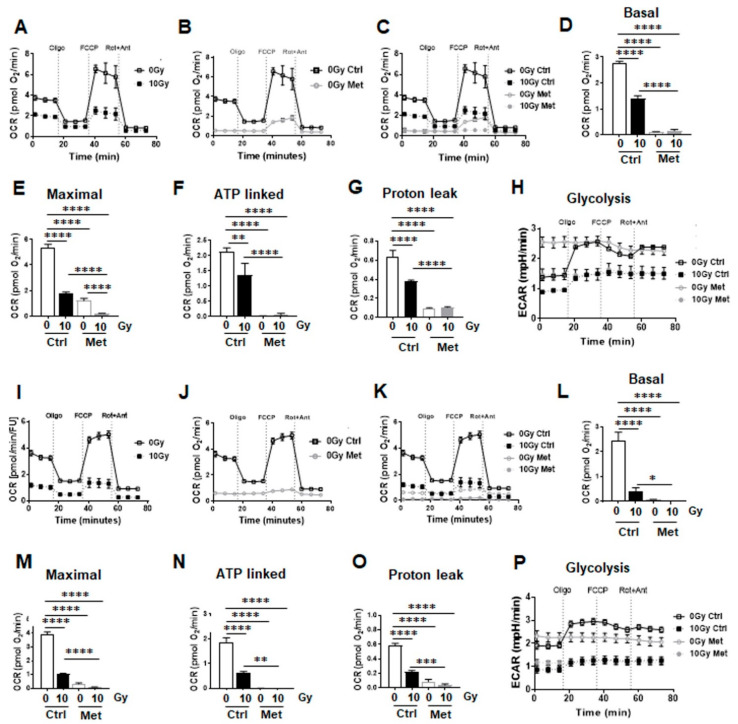
Metformin treatment reduces mitochondrial respiration in non-malignant epithelial lung cells. Extracellular flux analyses following (24 h after) metformin and RT treatment were performed in order to measure oxygen consumption rates (OCR) and extracellular acidification rates (ECAR) in BEAS-2B (**A**–**H**) and HSAEC1-KT (**I**–**P**) over time, while oligomycin (Oligo; 1 µM), FCCP (1 µM) and rotenone/Antimycin A (Rot/Ant; 0.5 µM) were added at indicated time points. (**A**,**I**) OCR levels of control and irradiated lung epithelial cells are shown. (**B**,**J**) OCR levels of control and metformin treated lung epithelial cells are shown. (**C**,**K**) Decreases in OCR following the combined treatment are shown. (**D**,**L**) Basal respiration, (**E**,**M**) maximal respiration, (**F**,**N**) ATP production, and (**G**,**O**) proton leak were depicted in separate bar diagrams. (**H**,**P**) Respective ECAR measurement over time are shown. Data were summarized as mean values ± SD (measured in 4–7 replicates each). One of 3 independent experiments per epithelial cell line with similar results is shown. *p*-values indicate: * *p* ≤ 0.05, ** *p* ≤ 0.01, *** *p* ≤ 0.001, **** *p* ≤ 0.0001 by one-way ANOVA with Tukey’s multiple comparison test.

**Figure 8 ijms-22-07064-f008:**
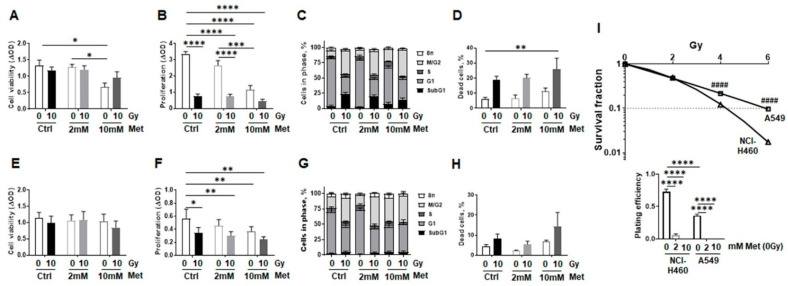
Metformin treatment reduces malignant lung epithelial cell proliferation and survival following RT. Non-small cell lung cancer cells (NSCLC; NCI-H460: human epithelial large cell lung carcinoma; (**A**–**D**) and A549: human alveolar basal epithelial adenocarcinoma; (**E**–**H**)) were cultured in normal growth media supplemented with metformin (2 and 10 mM) or vehicle control (Ctrl) 2 h prior radiation treatment with 0Gy or 10Gy. Cell viability (**A**,**E**) cellular proliferation (**B**,**F**), cell cycle distributions (**C**,**G**), and cell death induction (**D**,**H**) were analyzed 96 h after the treatments. (**A**,**E**) Cell viability measurements as determined by metabolic activity using the WST-1 reagent are summarized as mean values ±SEM of 4–5 experiments, with seven replicates each. (**B**,**F**) Cellular proliferation was analyzed by crystal violet staining. Data are presented as mean values ±SEM of 5 independent experiments. *p*-values indicate * *p* ≤ 0.05, ** *p* ≤ 0.01, *** *p* ≤ 0.001, **** *p* ≤ 0.0001 by two-way ANOVA with Tukey’s multiple comparison test. (**C**,**G**) Respective cell cycle analysis at that time point were estimated following Nicoletti staining. Data were summarized as mean values ±SEM of 5–7 independent experiments. (**D**,**H**) Total cell death levels were determined by propidium iodide exclusion staining using flow cytometry. Data are summarized as mean values of 3–5 independent experiments. *p*-value indicate ** *p* ≤ 0.01 by two-way ANOVA with Tukey’s multiple comparison test. (**I**) Clonogenic survival was evaluated in metformin treated (0, 2, 10 mM) and irradiated (0, 2, 4, 6 Gy) NCI-H460 and A549 cells, 10 days post treatment. Coomassie stained colonies were quantified and respective surviving fractions from 3 independent experiments (means ±SEM), plated in triplicates each, were depicted. As metformin treatment dramatically reduced the plating efficiency of both NSCLC cells in a concentration-dependent manner, no colonies could be quantified here. Statistics were analyzed by unpaired (two-tailed) *t*-tests, comparing NCI-H460 cells to corresponding A549 cells.

**Figure 9 ijms-22-07064-f009:**
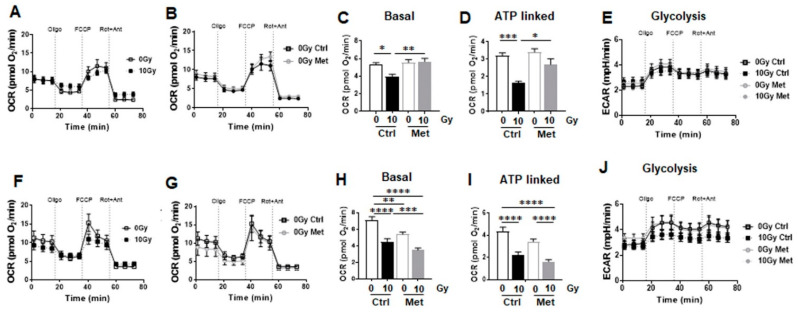
Metformin treatment does not impact on mitochondrial respiration in NSCLC, nor in combination with RT. Extracellular flux analyses following (24 h after) metformin and RT treatment were performed in order to measure oxygen consumption rates (OCR) and extracellular acidification rates (ECAR) in NCI-H460 (**A**–**E**) and A549 (**F**–**K**) over time, while oligomycin (Oligo; 1 µM), FCCP (2 µM) and rotenone/Antimycin A (Rot/Ant; 0.5 µM) were added at indicated time points. (**A**,**F**) OCR levels of control and irradiated malignant epithelial cells are shown. (**B**,**G**) OCR levels of control and metformin treated malignant epithelial cells are shown. (**C**,**H**) Basal respiration and (**D**,**I**) ATP-liked respiration were depicted in separate bar diagrams. (**E**,**K**) Respective ECAR measurement over time are shown. Data are summarized as mean values ±SEM of 3 independent measured in 4–7 replicates each. *p*-values indicate: * *p* ≤ 0.05 ** *p* ≤ 0.01, *** *p* ≤ 0.001, **** *p* ≤ 0.0001 by one-way ANOVA with Tukey’s multiple comparison test.

**Figure 10 ijms-22-07064-f010:**
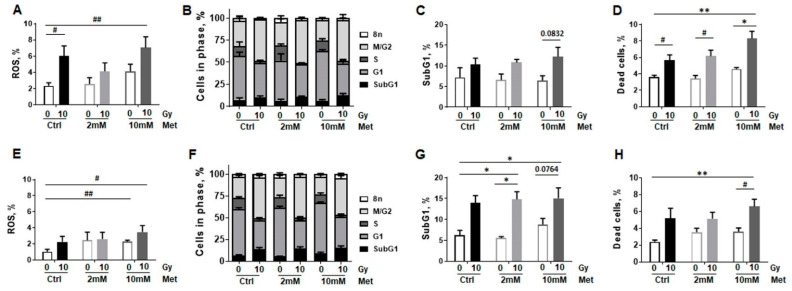
RT-induced cellular stress is increased upon combined metformin treatment in malignant lung epithelial cells. (**A**,**E**) NCI-H460 and A549 lung tumor cells were analyzed concerning reactive oxygen species (ROS) production at 24 h of post RT and metformin treatment using dihydroethidium (DHE) staining in combination with flow cytometry analyses. Data are shown as mean values ±SEM of 5–7 independent experiments. Statistics were analyzed by unpaired (two-tailed) *t*-tests, indicating: # *p* ≤ 0.05 and ## *p* ≤ 0.01. *p*-value indicates: ** *p* ≤ 0.01 by two-way ANOVA with post hoc Tukey’s multiple comparison test. (**B**,**F**) Respective cell cycle analysis at that time point were estimated following Nicoletti staining. Data were summarized as mean values ±SEM of 3 independent experiments. (**C**,**G**) Cell death rates were further determined by exclusion propidium iodide staining. Data show mean values ±SEM of 3 independent experiments. *p*-values indicate: * *p* ≤ 0.05, ** *p* ≤ 0.01 two-way ANOVA with Tukey’s multiple comparison test. Unpaired (two-tailed) *t*-tests are additionally indicated: # *p* ≤ 0.05. (**D**,**H**) Total cell death levels were determined by propidium iodide exclusion staining using flow cytometry. Data are summarized as mean values of 3–5 independent experiments. *p*-value indicate: * *p* ≤ 0.05, ** *p* ≤ 0.01 by two-way ANOVA with Tukey’s multiple comparison test. Unpaired (two-tailed) *t*-tests are additionally indicated: # *p* ≤ 0.05.

## Data Availability

Data are available from the corresponding author upon reasonable request.

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
