# Peer review of "Metformin Protects against Radiation-Induced Acute Effects by Limiting Senescence of Bronchial-Epithelial Cells"

_ijms, 2021, doi:10.3390/ijms22137064_

Round 1
Reviewer 1 Report
This is a well-written manuscript where the authors addressed how the problem of post radiation therapy lungs-damage could be mitigated. The COVID crisis has renewed scientific communities focus on various aspects of lung damages and in this context the findings of this research work will be an important addition. The authors used Metformin which is one of the most prescribed medicine and its toxicity and side-effects are well known. Because of the current importance of finding new therapeutic potentials for repairments of damaged lungs I recommend this work for acceptance to publication.
Experimental approaches are appropriate and adequate control experiments have been done. Statistical methods are adequate though the n=6 is the acceptable minimum number of animals in a study.
The authors are advised to add an extra paragraph explaining the rationale of using such high dosage of Metformin. They have used 125mg/g which seems very high considering the fact that human dosage for Metformin seldom exceeds 2000mg for a 60Kg individual. Therefore, an explanation is needed for selecting such high dosage amount. Also, as Metformin is typically given orally so the authors need to explain why they have decided to administer it though ip injection instead of oral gavage. Explaining these points will strengthen the paper.
Author Response
First, we would like to thank for the positive evaluation of our manuscript enabling us the re-submission of our revised manuscript. According to the reviewer’s suggestions, we addressed and corrected the critical points. Changes were emphasized within the manuscript using yellow color.
We absolutely agree with the reviewer’s suggestion about including an extra paragraph explaining the rationale of using such high dosage of metformin. This has been included now at the end of the discussion section (starting from line 724). According to the reviewer’s suggestion we also included the rationale for administering metformin though intraperitoneal injection instead of oral gavage. This has been included now in the Materials and Methods (Whole thorax irradiation (WTI)) section (starting from line 763).
Reviewer 2 Report
That is a fascinating manuscript describing well-designed and well-performed experiments on metformin use in the lung radioprotection. The study was performed on an animal model and using in vitro cultured human cell lines.
Overall - congratulations to Authors!
I have only some minor comments:
In the abstract, please specify that the preclinical model is an animal in origin – just for clarity.
Data presentation in Figures, particularly Figs description, need to be improved. Please check all Figures legend for correctness - e.g., in Fig. 1, control vs. metformin changes were described in the Results section as significant („Similar results were detected for mRNA expression levels of the immune-related SASP factors Ccl7, Ccl8, Cxcl1 (C-X-C motif chemokine 1) and Il6 compared to respective Ccl2 expressions of control and metformin treated WTI mice”). However, significances have not been depicted in Fig.
Figure 3 C – It seems that there is a significant difference between the number of sites of vascular leakage at baseline between CTR and Metformin groups. Thus, the difference regarding CTR vs. metformin comparison should be presented here in reference to the changes from baseline and not as the raw data (or maybe they are, not clear – please clarify). Furthermore, the differences are shown with four stars - not explained in the Fig. legend (“** p ≤ 0.0001”).
Part 2.2. The authors investigated two epithelial cell lines, i.e., BEAS-2B and HSAEC1-KT, describing them as “normal lung epithelial cells.” However, both lines were cloned and immortalized, thus do not reflect normal human epithelial cells. In my opinion, the “normal” term could be used only for primary epithelial cells obtained from healthy individuals’ airways, eg, using the air-liquid interface approach. Therefore, I would recommend using the term “epithelial cell line” instead of “normal epithelial cells”, or limit the use of “normal epithelial cells” to the context when these cell lines are compared to cancer cells. Again, in my opinion, immortalization might also have an impact on the study outcomes.
Results section 252: In both sentences: “While metformin treatment resulted in a slight increase of cells in G1 phase in a concentration dependent manner (48% at 0 mM to 53% at 10 mM) in non- irradiated cells. Metformin treatment increased the number of cells in G2/M phase in a concentration dependent manner (from 27% at 0 mM to 37% at 10 mM) upon irradiation with 15 Gy” - please provide whether the differences are significant. The same in line 420.
Figure 6 A and C – Fig. description, please clarify what A and C is? Are any significant differences in C ? – not depicted; however, in the main text is stated:” ….metformin…. reduced ROS levels in irradiated HSAEC1-KT cells (Figure 6C)”.
Methods: mice received intraperitoneal injection of 125 mg/g metformin, which is a much higher dose than routinely applied for humans (av. 1000-2000 mg per day). Could the Authors comment about their opinion on whether the lower doses of metformin used in medicine might also be beneficial in lung radioprotection? Are any reliable data on that issue in humans? – topic is particularly interesting for clinicians.
Author Response
First, we would like to thank for the positive evaluation of our manuscript enabling us the re-submission of our revised manuscript. According to the reviewer’s suggestions, we addressed and corrected the critical points. Changes were emphasized within the manuscript using yellow color.
Abstract: According to the reviewer’s suggestion we specified that the preclinical model is a mouse model.
Data presentation in Figures: We critically checked the presented data, the respective results text and the Figures legends for correctness and adjusted corresponding statements.
Figure 3C: According to the reviewer’s suggestion we included the significant difference between the number of sites of vascular leakage at baseline between Ctrl and Metformin groups. We apologize for the lack of explanation for four stars; this was a typing error (of two missing stars) and has been corrected now.
Part 2.2.: We absolutely agree with the reviewer’s suggestion that both cell lines used are “only” model approaches for normal lung epithelial cells. According to the reviewer’s recommendation we limited the use of “normal epithelial cells” to a minimum and only to the context when these cell lines are compared to cancer cells. Apart from that we used the term “epithelial cell line” or “non-malignant epithelial lung cells” instead of “normal epithelial cells”.
Results section:
According to the reviewer’s suggestion we investigated whether the re-checked if indicated differences were significant. For clarification, we included now additional diagrams depicting respective G1 and G2/M levels (96 post RT) of all four cell lines investigated in the new additional Supplemental Figures S1A,B (BEAS-2B and HSAEC1-KT) and S1C,D (NCI H460 and A549) and critically checked the respective results text for correctness.
Figure 6A and C: We apologize for the unclear description and improved the description as well as the statement within the results section. At this stage we are happy to include now additional biological replicates (for ROS measurements; Figure 6A: BEAS-2B, Figure 6C: HSAEC1-KT and Figure 10A: NCI-H460, Figure 10E: A549 as well as Supplemental Figure S2 (former Figure S1) that were prepared in advance for a revision and now confirmed our findings. We re-performed statistical analyses and consistently adjusted the text within the result section.
Methods (and Discussion): We included now an extra paragraph explaining the rationale of using such high dosage of metformin at the end of the discussion section (“limitations” starting from line 724). We further included the rationale for administering metformin though intraperitoneal injection instead of oral gavage. This has been included now in the Materials and Methods (Whole thorax irradiation (WTI)) section (starting from line 763).